# Evaluation of a 3-Item Health Index in Predicting Mortality Risk: A 12-Year Follow-Up Study

**DOI:** 10.3390/diagnostics13172801

**Published:** 2023-08-29

**Authors:** Silvin P. Knight, Mark Ward, Eoin Duggan, Feng Xue, Rose Anne Kenny, Roman Romero-Ortuno

**Affiliations:** 1The Irish Longitudinal Study on Ageing (TILDA), School of Medicine, Trinity College Dublin, D02 R590 Dublin, Ireland; 2Discipline of Medical Gerontology, School of Medicine, Trinity College Dublin, D02 R590 Dublin, Ireland; 3Mercer’s Institute for Successful Ageing (MISA), St. James’s Hospital, D08 NHY1 Dublin, Ireland; 4Global Brain Health Institute, Trinity College Dublin, D02 PN40 Dublin, Ireland

**Keywords:** 3-item health index, sample entropy, SART, gait speed, mortality, frailty, healthcare computer application, TILDA, MATLAB health index app

## Abstract

This study was carried out using a large cohort (*N* = 4265; 416 deceased) of older, community-dwelling adults from The Irish Longitudinal Study on Ageing (TILDA). The study compared the performance of a new 3-item health index (HI) with two existing measures, the 32-item frailty index (FI) and the frailty phenotype (FP), in predicting mortality risk. The HI was based on the objective measurement of resting-state systolic blood pressure sample entropy, sustained attention reaction time performance, and usual gait speed. Mortality data from a 12-year follow up period were analyzed using Cox proportional regression. All data processing was performed using MATLAB and statistical analysis using STATA 15.1. The HI showed good discriminatory power (AUC = 0.68) for all-cause mortality, similar to FI (AUC = 0.68) and superior to FP (AUC = 0.60). The HI classified participants into Low-Risk (84%), Medium-Risk (15%), and High-Risk (1%) groups, with the High-Risk group showing a significant hazard ratio (HR) of 5.91 in the unadjusted model and 2.06 in the fully adjusted model. The HI also exhibited superior predictive performance for cardiovascular and respiratory deaths (AUC = 0.74), compared with FI (AUC = 0.70) and FP (AUC = 0.64). The HI High-Risk group had the highest HR (15.10 in the unadjusted and 5.61 in the fully adjusted models) for cardiovascular and respiratory mortality. The HI remained a significant predictor of mortality even after comprehensively adjusting for confounding variables. These findings demonstrate the effectiveness of the 3-item HI in predicting 12-year mortality risk across different causes of death. The HI performed similarly to FI and FP for all-cause mortality but outperformed them in predicting cardiovascular and respiratory deaths. Its ability to classify individuals into risk groups offers a practical approach for clinicians and researchers. Additionally, the development of a user-friendly MATLAB App facilitates its implementation in clinical settings. Subject to external validation in clinical research settings, the HI can be more useful than existing frailty measures in the prediction of cardio-respiratory risk.

## 1. Introduction

In 2015, approximately 8.5% of the global population was aged 65 years or older. This percentage is expected to rise to 13% by 2030 and further to 16.7% by 2050, implying an average annual growth of 27.1 million older individuals from 2015 to 2050 [1]. Aging is a complex process that involves various physiological, cognitive, and functional changes. The aging process is associated with an increased risk of morbidity and mortality, leading to significant healthcare costs and diminished quality of life. Identifying individuals at a higher risk of morbidity and mortality is critical for implementing appropriate interventions that can improve health outcomes and reduce healthcare costs [2]. Frailty is a geriatric syndrome that reflects increased vulnerability to adverse health outcomes from stressors, including falls, hospitalization, institutionalization, and mortality [3,4,5,6]. Yet, frailty measures have limitations, including reliance on self-report, necessitating new approaches to improve their predictive ability [7].

In this study, we developed and tested a new health index (HI) that addresses some of the limitations of existing frailty measures by incorporating three objective items: resting-state systolic blood pressure (sBP) sample entropy (SampEn), sustained attention reaction time (SART) performance, and usual gait speed (UGS). These objective and quantitative measures were chosen based on their sensitivity to changes in cardiovascular, neurocognitive, and locomotor status, respectively, and their potential to provide a more comprehensive assessment of an individual’s mortality risk based on three combined critical, core physiological systems. 

Resting state sBP SampEn is a measure of blood pressure irregularity that reflects dysregulation in autonomic nervous system function and cardiovascular health. Entropy, in general, is a measure of irregularity or unpredictability in physiological signals. Lower entropy values indicate more periodic and predictable data, while higher entropy values indicate more irregular and unpredictable data. Previous work has shown elevated blood pressure SampEn, in both population-level and clinical cohorts, to be associated with: increased risk of mortality; worse pre-disability frailty status; worse longitudinal cognitive performance; and increased risk of future falls, syncope, and fear of falling [8,9,10,11,12].

SART is a cognitive test designed to measure sustained attention, a fundamental executive function necessary for tasks requiring supervision over time [13]. The state of sustained attention arises from the interplay of two distinct subsystems: vigilance and arousal, also known as alertness [14,15]. Vigilant attention allows a person to detect subtle changes in their environment over extended periods of time [15,16], and neuroimaging studies have demonstrated that this relies on a network of cortical areas, including the cingulate gyrus, prefrontal cortex, and inferior parietal lobe [17,18]. Maintaining an adequate level of arousal is necessary in order to detect target stimuli, and electrophysiology and functional neuroimaging studies have demonstrated that arousal is activated through a subcortical network which includes the thalamus and noradrenergic brainstem structures [19,20]. The SART task assesses sustained attention by measuring commission errors (responding to non-target stimuli) and omission errors (failure to respond to target stimuli) along with reaction time [21,22]. Impairments in sustained attention have been linked to frailty and falls efficacy in older adults [22,23]. By including SART performance as a component of the HI, we aimed to capture neurocognitive health and its influence on mortality risk.

Usual gait speed (UGS) is a measure of physical function and mobility that reflects an individual’s general fitness [24]. Faster UGS is associated with meeting occupational demands in younger adults [25], while slower UGS is associated with functional decline and increased morbidity in older adults [26,27]. UGS is commonly assessed in clinical practice and has well-established associations with age, physical function, and frailty [28,29,30]. By incorporating UGS as a component of the HI, we evaluated its impact on overall health status and mortality risk.

The 3-item HI is a continuous measure ranging from 0 (lowest risk) to 1 (highest risk) and demonstrates a relatively normal distribution. Unlike many existing frailty measures, which have limitations related to the count-based, largely self-reported approach, and predefined criteria [3,4,31], the HI relies on objective and continuous health measures and a data-driven quantification of their combined dysregulation. Notably, other researchers previously proposed mortality-specific indices, employing models based on 10 to 14 predictor variables, which significantly predicted mortality [32,33,34]. Yet, these models continue to rely on self-reported data, and we therefore hypothesized that the new 3-item HI would predict mortality more accurately than existing measures. Moreover, to facilitate its use in clinical research settings, an open and easily accessible MATLAB application was developed and is readily downloadable.

Specifically, this study sought to address the following research questions:How does the newly developed 3-item HI, integrating objective physiological markers such as resting-state sBP SampEn, SART performance, and UGS, compare with frailty measures such as the 32-item FI and FP in predicting all-cause mortality risk?To what extent does the 3-item HI compare with the above frailty measures in identifying individuals at heightened risk of specific-cause mortality (e.g., cardio-respiratory)?

## 2. Materials and Methods

### 2.1. Design, Setting, and Participants

This study was conducted as part of an ongoing prospective cohort study focusing on older adults residing in the community in Ireland, known as The Irish Longitudinal Study on Ageing (TILDA) [35,36]. TILDA aims to gather information on the health, economic, and social circumstances of individuals aged 50 years and above. The baseline assessment, referred to as Wave 1, occurred between October 2009 and February 2011, with a total of 8507 participants (primarily assessed in 2010). The study’s sample was drawn from the Irish Geodirectory, a comprehensive and current database containing all residential addresses across the Republic of Ireland, compiled by ‘An Post’ (the Irish Postal Service) and Ordnance Survey Ireland. The initial selection of addresses for the sample was carried out using the RANSAM sampling procedure [37], a multi-stage probability sampling method developed by the Economic and Social Research Institute [36]. Subsequent data collection took place approximately every two years across five longitudinal waves: Wave 2 (February 2012 to March 2013), Wave 3 (March 2014 to December 2015), Wave 4 (January to December 2016), Wave 5 (January to December 2018), and Wave 6 (January 2021 to December 2021). Of note, for COVID-19 pandemic reasons, Wave 6 data only included the computer assisted telephone interview (CATI).

The health assessments in Waves 1 and 3 were comprehensive and carried out at a dedicated health assessment center, while Waves 2, 4, and 5 involved non-health center assessments. A detailed description of the entire cohort profile has been previously provided [35,36]. The research received ethical approval for each wave from the Faculty of Health Sciences Research Ethics Committee at Trinity College Dublin, Dublin, Ireland, and all participants provided written informed consent. The study adhered to the principles outlined in the Declaration of Helsinki.

### 2.2. Blood Pressure Data

During the initial health assessment (Wave 1), blood pressure waveforms were continuously measured at a rate of 200 Hz using a Finometer MIDI device (Finapres Medical Systems BV, Amsterdam, The Netherlands). These measurements were recorded using a 12-bit resolution analogue-to-digital converter. The assessments took place in a comfortably lit room with an ambient temperature of 21 to 23 °C. Participants were instructed to lie in a supine position, and after a period of stabilization, a minimum of five minutes of data were collected. For the main analyses conducted in this study, the data from the final minute of supine rest (referred to as the resting state or RS) was used. This selection aimed to maximize data stationarity by capturing a period of natural stability as much as possible. It has also been previously shown that 1 min of RS BP data are adequate for calculating BP SampEn using the methods outlined herein [8]. Entropy analysis was performed on the sBP data, using freely available MATLAB code [38]. The computation of SampEn involved specific algorithms, which have been extensively described in previous research [39]. Here, we provide a concise overview of these algorithms.

For a given timeseries of length *N*, $B_{i}^{m}(r)$ represents the count of template vectors of length *m*, denoted as $x_{m}(j)$, that are similar to $x_{m}(i)$ (within a threshold of *r** the standard deviation (SD) of the timeseries). This count is divided by *N* − *m* − 1, where *j* ranges from 1 to *N − m*, with *j* not equal to *i* (to exclude self-matches). The average of $B_{i}^{m}(r)$ across all *i* is defined as $B^{m}\left(r\right)$ and is calculated as:(1)$$B^{m}\left(r\right)=\frac{1}{N-m}\sum_{i=1}^{N-m}B_{i}^{m}\left(r\right).$$

Similarly, $A_{i}^{m}(r)$ represents the count of template vectors of length *m* + 1, denoted as $x_{m+1}(j)$, that are similar to $x_{m+1}(i)$ (within a threshold of *r*). This count is divided by *N* − *m* − 1, where *j* ranges from 1 to *N − m*, with *j* ≠ *i*. The average $A_{i}^{m}(r)$ across all *i* is defined as $A^{m}\left(r\right)$ and is calculated as:(2)$$A^{m}\left(r\right)=\frac{1}{N-m}\sum_{i=1}^{N-m}A_{i}^{m}(r).$$

SampEn was then calculated as
(3)$$SampEn\left(m,r,N\right)=-ln\left(\frac{A^{m}\left(r\right)}{B^{m}\left(r\right)}\right).$$

In simpler terms, SampEn (as calculated in Equation (3)) captures the likelihood of finding similar patterns in the data, considering different lengths of sequences. When the ratio of longer patterns ($A^{m}\left(r\right)$: Equation (2)) to shorter patterns ($B^{m}\left(r\right)$: Equation (1)) is smaller, it suggests greater complexity and unpredictability in the data, leading to a higher SampEn value. Conversely, a larger ratio indicates more regularity and results in a lower SampEn value.

In this study, *m* (the embedding dimension; the length of the data segment being compared) was set to 1 and *r* (the similarity criterion) was set to 0.4, as per previous work investigating the optimal *m* and *r* for use in the prediction of mortality [8].

### 2.3. SART Data

The Sustained Attention to Response Task (SART) is a computerized continuous performance reaction time (RT) task [13]. It involves participants responding to a sequential stream of digits from 1 to 9 (GO trials) but refraining from responding when the digit 3 appears (NO-GO trials).

During the SART test, each digit is displayed for 300 ms, followed by an 800 ms interval before the next digit appears. The cycle of digits 1 to 9 is repeated 23 times, resulting in a total of 207 trials. The test duration is approximately 4 min. Participants are instructed to press a designated key on the keyboard as quickly as possible when presented with each digit. The RT is automatically recorded using Presentation version 16.5 software. In the SART task, when participants respond correctly and refrain from pressing the key for the digit 3, the RT is recorded as zero (RT = 0). In an ideal scenario, there would be 184 non-null values representing the RTs when participants are supposed to respond (8 possible digits per cycle × 23 cycles), and 23 null values for the trials when participants are not expected to respond. However, in practice, participants may lose attention during the test, leading to errors.

The data collected from the SART task can reveal two types of mistakes. Commission errors occur when participants respond to NO-GO trials, indicating lapses in sustained attention. Omission errors, on the other hand, happen when participants fail to respond to GO trials, indicating a temporary disengagement from the task due to lapses in attention [22]. As adapted from previous studies using SART data, a single ‘bad performance’ was herein defined as a trial where the participant committed at least 2 mistakes out of 9 possible actions [40,41]. The number of bad performances (NBPs) in the SART task was then taken as the sum of bad performances across all 23 cycles (i.e., potential minimum NBP = 0 and maximum NBP = 23). For this study, the SART data used was also obtained from Wave 1 of TILDA.

### 2.4. Gait Data

Gait speed was evaluated at Wave 1 of TILDA using a 4.88 m long GAITRite (CIR Systems Inc., Franklin, NJ, USA) computerized pressure-sensing walkway [42,43]. Participants were instructed to walk at their usual pace and performed two walks on the walkway. The walks started 2.5 m before the walkway and ended 2.0 m after it. To calculate the usual gait speed (UGS), the measured speed in centimeters per second (cm/s) was averaged between the two walks. It should be noted that the calculation excluded the acceleration and deceleration phases of walking, focusing only on the steady-state walking speed.

### 2.5. Three-Item HI

The 3-item HI was mathematically formulated based on normalization and combination of sBP SampEn, NBP in the SART task, and UGS. The formulation involved normalizing each variable to the full range of values observed in the TILDA dataset, enabling standardized comparisons and interpretations of the HI scores.

The resulting HI provides a composite measure of health based on these three normalized variables, where higher values indicate worse health status. The normalization and calculation of the 3-item HI were implemented using MATLAB. These methods allow for consistent and standardized calculation of the HI, facilitating its application in clinical research settings. Appendix B details the normalization process and 3-item HI derivation.

The chosen methodology was underpinned by deliberate considerations to address the research objectives effectively while leveraging the large TILDA dataset to derive HI scores normalized to the large cohort. Notably, the formulation of the 3-item HI aimed to serve as a concise yet comprehensive measure of health, aggregating key objective variables, sBP SampEn, NBP in the SART task, and UGS, that independently were previously associated with negative health outcomes, including mortality risk [8,9,10,11,12,22,23,26,27,28,29,30].

### 2.6. Three-Item HI Risk Groups

To define risk groups based on HI scores, a data-driven approach was employed. Multiple univariate Cox models were utilized to split the data into groups at various HI values, ranging from 0.2 to 0.65 in increments of 0.025. The hazard ratios (HRs) and *p*-values for the prediction of all-cause mortality were then plotted to identify the optimal cut-off that maximized the HR and minimized the *p*-value, enabling the differentiation of ‘Low-Risk’ individuals from the remaining cohort. This process was repeated to further separate ‘Medium-Risk’ and ‘High-Risk’ individuals from each other and the Low-Risk group. The plots used to determine the cut-off values are provided in Appendix B.

### 2.7. Mortality Data

The official records of participants’ deaths were used to determine their date and cause (cardiovascular, respiratory, cancer, or other) of death, which was then connected to their TILDA survey and health assessment data. This linking process was carried out for individuals who passed away from April 2010 to January 2022. More information about the specific procedures for linking the data can be found in a separate description [44].

### 2.8. Frailty Measures

This study compared the effectiveness of the new 3-item HI with two established methods of measuring frailty: the FP originally operationalized by Fried et al. [3] and the 32-item FI [4], both of which were previously operationalized in TILDA.

To determine the FP, we followed the methodology proposed by Fried et al. [3]. Detailed information has been previously described [22,45,46]. In summary, the FP was assessed using specific cut-off points tailored to the population. These cut-off points were based on variations in measurements of weakness (adjusted grip strength using a dynamometer on the dominant hand, adjusted for sex and body mass index), physical activity (sex-adjusted kilocalories from the International Physical Activity Questionnaire—Short Form [47]), and walking speed (adjusted for sex and height, measured in centimeters per second using the GAITRite portable walkway). Weight loss was determined by asking participants if they had unintentionally lost 10 pounds (4.5 kg) or more in the past year. Exhaustion was identified by two items from the Centre for Epidemiological Studies Depression (CESD) scale [48]. Participants were asked how frequently they experienced feelings of “not being able to get going” and “feeling that everything they did was an effort.” A response of “moderate amount/all of the time” to either question was considered as indicating “exhaustion”. As per previous literature [3,22,45,46], ‘frail’ was defined as having 3 or more criteria (from low grip strength, low physical activity, low gait speed, unintentional weight loss, or self-reported exhaustion), ‘pre-frail’ as having 1 or 2 criteria, and ‘non-frail’ as having none.

Additionally, we constructed a 32-item FI using self-reported health measures obtained during the initial wave of the TILDA study [49]. The selection of deficits for this index followed the standard requirements for an FI [50]. These deficits included symptoms, signs, diseases, or disabilities associated with aging and adverse outcomes, present in at least 1% of the population, covering various organ systems, and having less than 5% missing data [49]. The components of this 32-item FI and the scoring scheme for each item can be found in Appendix D. Previous research has suggested that FI variables can be either dichotomous or ordinal, with minimal impact on the predictive ability of the FI [51]. In line with previous literature [52], the following cut-offs were applied for the definition of the three FI states: FI < 0.10, ‘non-frail’; 0.10 ≤ FI < 0.25, ‘pre-frail’; and FI ≥ 0.25, ‘frail’.

### 2.9. Covariates

Within the TILDA survey, several self-reported measures were collected at Wave 1 and utilized as covariates in the fully adjusted models presented in this study. These measures included age, sex, educational attainment, the presence of various cardiovascular conditions (such as angina, heart murmur, abnormal heart rhythm, heart attack, stroke, or transient ischemic attack (TIA)), diabetes, alcohol consumption habits assessed using the CAGE questionnaire [53], smoking history, and the use of antihypertensive medications. The antihypertensive medication use was coded using the Anatomical Therapeutic Chemical Classification (ATC), which categorized medications into classes such as antihypertensive medications (ATC C02), diuretics (ATC C03), β-blockers (ATC C07), calcium channel blockers (ATC C08), and renin-angiotensin system agents (ATC C09).

Furthermore, during the Wave 1 health center assessment, anthropometric measurements were taken, including height (measured to the nearest 0.01 m using Seca 240 Stadiometer (Seca Ltd., Birmingham, UK)) and weight (measured to the nearest 0.1 kg using Seca 861 Electronic Scales (Seca Ltd., Birmingham, UK)). Body mass index (BMI) was calculated using the formula weight [kg]/(height [m])^2^. To account for the likely non-linear relationship between BMI and mortality [54], BMI was categorized into four groups based on the World Health Organization (WHO) guidelines [55]: ‘underweight/normal’ (BMI < 25), ‘overweight’ (25 ≤ BMI < 30), ‘obese’ (30 ≤ BMI < 35), and ‘morbidly obese’ (BMI ≥ 35).

### 2.10. Statistical Analysis

The statistical analysis was conducted using STATA 15.1 (StataCorp, College Station, TX, USA). Visual examination was conducted on the distributions of continuous variables using histograms and distributional diagnostic plots and tested using the skewness kurtosis test for normality. Summary statistics of the overall cohort and HI groups (‘Low-Risk’, ‘Medium-Risk’, and ‘High-Risk’) were calculated as mean and SD for continuous normally distributed variables, while non-normally distributed continuous variables were summarized using the median and interquartile range (IQR). Proportions were presented as both counts and percentages.

Cox proportional hazards regression models were employed to estimate hazard ratios (HRs) with 95% confidence intervals (CIs) for the association between the 3-item HI, FI, and FP with both all-cause and specific-cause mortality. Nonparametric receiver operating characteristic (ROC) analysis was also performed and the area under the curve (AUC) calculated. For specific cause analysis, we chose to focus on cardiovascular and respiratory deaths and combined these groups into a single category. This decision was made due to a relatively low overall number of deaths and the overlapping nature of cardiovascular and respiratory conditions, as they share common risk factors and often exhibit similar pathological mechanisms [56]. Participants who were lost to follow-up were right-censored at the end of the follow-up period (31 January 2022). The purpose of this approach was to appropriately handle individuals who did not die within the study period.

In order to assess the independent predictive power of the 3-item HI, we used two models: (i) an unadjusted model and (ii) a fully adjusted model. The adjustment aimed to account for potential confounding effects that may arise from covariates commonly associated, clinically and/or epidemiologically, with mortality risk. These covariates included age, sex, education, BMI, antihypertensive medication use, diabetes, number of cardiovascular conditions, smoking status, and alcohol consumption.

### 2.11. MATLAB App and Code

To facilitate the calculation of the new 3-item HI, a MATLAB app was designed and developed. The app is available on GitHub (https://github.com/SilvinPKnight/3ItemHealthIndexCalculator, accessed on 18 July 2023) (Appendix A) and can be installed into MATLAB for easy utilization. The MATLAB App was designed to streamline the process of loading the required data for the calculation of the HI. Specifically, the app allows the user to load the following data: raw beat-to-beat (BtB) sBP, raw SART reaction times, and raw gait speed data. Once the required data are loaded into the app, Wave 1 TILDA values embedded in the code serve as reference for deriving the HI scores. The MATLAB app and code provide a user-friendly interface for clinical researchers to calculate the 3-item HI efficiently and accurately. Further details on the functionality, installation instructions, and usage guidelines of the MATLAB App and code are provided in Appendix E.

## 3. Results

### Participant Characteristics

Figure 1 presents a flow diagram illustrating the participant exclusion process for this study. The initial cohort comprised 8507 individuals at Wave 1. After excluding 332 participants under the age of 50 and 3140 individuals with no health center assessment, the final sample included 5035 participants. Further exclusions were made due to missing data, resulting in a final analytical sample size of 4265 individuals. Specifically, 86 participants were excluded due to missing gait data, 161 due to missing SART data, 436 due to missing BP data, 86 due to missing FP data, and 1 participant due to missing educational attainment data. Of the 4265 participants, 416 had passed away during the 12-year study period. Among the deceased individuals, 115 had died from cardiovascular conditions, 30 from respiratory issues, 190 from cancer, and 81 from other causes. To enhance the statistical power and focus of the analysis, we combined cardiovascular and respiratory deaths, resulting in a pooled group of 145 individuals who had died from these causes.

Table 1 provides a comprehensive overview of the demographic characteristics of the entire cohort, along with the distribution within each risk group of the 3-item HI. The study cohort comprised a total of 4265 participants, with a mean age of 61.6 years (SD = 8.2). Among the participants, 54% were female. Based on the analysis of HI scores, the cohort was divided into three risk groups: Low-Risk, Medium-Risk, and High-Risk. The optimal cut-off value of 0.45 yielded a Low-Risk group consisting of 3604 individuals out of the total 4265 (84.5%). Furthermore, a second cut-off value of 0.65 separated the cohort into a Medium-Risk group comprising 625 (14.7%) individuals and a High-Risk group consisting of 36 (0.8%) individuals (see Appendix B for plots used to derive cut-offs).

Several notable differences were observed across the risk groups for various demographic variables. First, the mortality status differed significantly among the groups, with the High-Risk group exhibiting the highest percentage of deceased individuals (36%), followed by the Medium-Risk (22%) and the Low-Risk (7%) groups. Additional differences were observed, in the expected directions, for the 32-item FI, FP, diabetes, number of cardiovascular diseases, antihypertensive medication use, and demographic variables. Table 1 also highlights that those at higher risk by the 3-item HI experienced more difficulties with activities of daily living (ADLs) and instrumental activities of daily living (IADLs).

Figure 2 illustrates the all-cause mortality ROC curves and survival plots for the 3-item HI, the 32-item FI, and the FP. The AUC values are reported for each measure. Table 2 presents the results of univariate and multivariate Cox proportional regression analyses, examining the association between each health measure and all-cause mortality risk. The reference group for the 3-item HI consisted of individuals classified as Low-Risk, while the reference group for the 32-item FI and FP comprised individuals classified as non-frail. For the 3-item HI, in the univariate analysis, both the Medium-Risk group (HR = 3.27, 95% CIs = 2.66, 4.01) and the High-Risk group (HR = 5.91, 95% CIs = 3.38, 10.34) exhibited significantly higher HRs compared with the Low-Risk group for all-cause mortality. These associations remained significant in the multivariate analysis, with the Medium-Risk group (HR = 1.75, 95% CIs = 1.40, 2.19) and the High-Risk group (HR = 2.06, 95% CIs = 1.16, 3.65) showing increased HRs. The 3-item HI demonstrated a moderate discriminatory power, with an AUC of 0.68 for predicting all-cause mortality.

Similar patterns were observed for the 32-item FI and the FP. In both measures, the pre-frail group (32-item FI: HR = 2.15, 95% CIs = 1.72, 2.69; FP: HR = 2.06, 95% CIs = 1.69, 2.52) and frail group (32-item FI: HR = 4.31, 95% CIs = 3.36, 5.53; FP: HR = 6.11, 95% CIs = 3.84, 9.71) showed significantly higher HRs compared with the non-frail group for all-cause mortality in the univariate analysis. These associations mostly remained significant (with the exception of pre-frail by FI) in the multivariate analysis (Table 2). The 32-item FI showed similar discriminatory performance for predicting all-cause mortality to the 3-item HI, with AUC value of 0.68; while the FP had lower discriminatory performance, with an AUC of 0.60.

Figure 3 illustrates the cardio-respiratory mortality ROC curves and survival plots for the 3-item HI, the 32-item FI, and the FP. Table 3 presents the results of univariate and multivariate Cox proportional regression analyses, examining the association between each health measure and cardio-respiratory mortality risk. For the 3-item HI, in the univariate analysis, both the Medium-Risk group (HR = 4.32, 95% CIs = 3.06, 6.09) and the High-Risk group (HR = 15.10, 95% CIs = 7.79, 29.28) exhibited significantly higher HRs compared with the Low-Risk group. These associations remained significant in the multivariate analysis, with the Medium-Risk group (HR = 2.17, 95% CIs = 1.47, 3.19) and the High-Risk group (HR = 5.61, 95% CIs = 2.84, 11.05) showing increased HRs. The AUC for the 3-item HI in predicting combined cardiovascular and respiratory deaths was 0.74.

For the 32-item FI (cardio-respiratory mortality, Table 3), both the pre-frail group and the frail group showed significantly higher HRs compared with the non-frail group in the univariate analysis. In the multivariate analysis, the pre-frail group did not exhibit a significant association, while the frail group did. The 32-item FI demonstrated an AUC of 0.70 for predicting combined cardiovascular and respiratory deaths. Similarly, for the FP, the pre-frail group and the frail group exhibited significantly higher HRs compared with the non-frail group in the univariate analysis. These associations remained significant in the multivariate analysis. The FP had an AUC of 0.64 for predicting combined cardiovascular and respiratory deaths.

## 4. Discussion

The present study compared the performance of a new 3-item Health Index (HI) with two existing frailty measures, the 32-item FI and the FP, in predicting 12-year mortality risk in TILDA. As shown before, existing frailty measures were significantly predictive of mortality [57,58]. However, the new 3-item HI, which was solely based on objective measures across cardiovascular, neurocognitive, and locomotor domains, showed the best discrimination at predicting cardio-respiratory mortality. This suggests that, subject to future external validation in clinical research settings, this new tool can be more useful than existing frailty measures in the prediction of cardiovascular and respiratory risk.

In terms of all-cause mortality, the discriminatory abilities of the 3-item HI, FI, and FP were comparable, with similar AUC values. Comparable HR values were also observed between respective groups for all-cause mortality. These findings are consistent with previous studies, including a 2022 meta-analysis of 58 previous studies (pooled *N* = 1,852,951, pooled number of deaths = 145,276) by Peng et al., which reported pooled HRs for specific-cause and all-cause mortality risk associated with frailty status (assessed using FI, FP, or FRAIL scale). For example, the meta-analysis reported pooled HRs of 1.42 (36 studies) for pre-frail individuals and 2.40 (48 studies) for frail individuals for the prediction of all-cause mortality, compared with the non-frail [57]. It is worth noting that other previously reported mortality-specific predictor models outperform both the 3-item HI and the frailty measures concerning all-cause mortality. Notably, reported AUCs of 0.774 (TILDA; Ireland) [32], 0.859 (UK; English Longitudinal Study of Ageing (ELSA)) [33], and 0.82 (USA; Health and Retirement Study (HRS)) [34] have been achieved by these indices. However, it should be acknowledged that these indices rely on 10 to 14 self-reported variables, which were specifically derived from a wide pool of self-report variables (41 to 67) in two of the studies to optimize mortality prediction [32,34]. Consequently, direct comparisons to the present study are challenging, as our aim was to develop a more objective, data-driven measure. Of the three measures that make up the HI, two (sBP SampEn [8] and gait speed [59]) have previously been associated with increased risk of mortality; however, since the reported results are ‘per unit’ of SampEn or gait speed, meaningful direct comparison of the individual performance of these components of the HI with the full 3-item HI is not possible. Although there is a lack of previous studies investigating the predictive associations between SART performance and mortality, it is known that impaired neurocognitive performance is associated with higher mortality risk [60,61].

Notably, when examining the prediction of combined cardio-respiratory mortality, the new 3-item HI demonstrated superior performance compared with the existing frailty measures. The AUC for the 3-item HI was 0.74, while for the FI and FP, the AUC values were 0.70 and 0.64, respectively. Furthermore, the HRs for the 3-item HI were consistently higher than those for the frailty measures. The HR for Medium-Risk individuals assessed using the 3-item HI was 2.17 in fully adjusted models, and for High-Risk individuals, it was 5.61. In contrast, the HR for pre-frail individuals assessed using the FI was 0.91, and for frail individuals, it was 1.73. Similarly, the HR for pre-frail individuals assessed using the FP was 1.72, and for frail individuals, it was 3.31. The HRs associated with the 3-item HI in the current study were also notably higher than those reported by Peng et al. for both cardiovascular (pre-frail: pooled HR = 1.63 (12 studies); frail: pooled HR = 2.64 (13 studies)) and respiratory (pre-frail: pooled HR = 2.16 (4 studies); frail: pooled HR = 4.91 (5 studies)) mortality risk [57].

The results of our study, in conjunction with previous research, suggest that while frailty measures such as the FI and FP remain valuable for assessing all-cause mortality risk, the new 3-item HI holds particular promise for identifying individuals at heightened risk of combined cardiovascular and respiratory mortality. The incorporation of objective multisystem physiological markers in the 3-item HI may provide a more direct assessment of physical health status. Further research is warranted to externally validate and refine the 3-item HI and explore its applicability in diverse populations and clinical settings. Ultimately, these efforts can contribute to more effective risk stratification and interventions aimed at reducing mortality associated with cardiovascular and respiratory diseases. In particular, the objective nature of the components of the 3-item HI, coupled with its better prediction of cardio-pulmonary mortality, make the 3-item HI a good candidate tool for the assessment of risk in real healthcare scenarios (e.g., pre-operative risk assessment), subject to clinical research and validation in these settings.

Figure A1 offers insights into the robustness of the HI measure’s performance across a range of cut-off values. Notably, the plot demonstrates that various cut-offs produce similar statistical results in the fully controlled models. This finding enhances confidence in the HI measure’s effectiveness and highlights its potential as a versatile tool for assessing mortality risk. The consistency of outcomes across a spectrum of cut-offs underscores the measure’s stability and lends further support to its potential applicability in different clinical contexts. While this derivation of the optimal risk cut-off groups contributes to the understanding of the HI measure’s performance, it also emphasizes the need for continued research in refining the optimal cut-off value, potentially incorporating domain-specific considerations. Further exploration is warranted to ascertain whether specific population characteristics or health conditions can influence the choice of cut-off and enhance the measure’s predictive precision. Additionally, investigating the longitudinal stability of the chosen cut-off values and their generalizability to diverse populations would provide a comprehensive perspective on the HI measure’s utility, as has been previously performed with frailty measures [62,63].

The advantages of the 3-item HI may extend beyond its predictive capabilities. The FP relies on count-based measures, which may not fully capture the nuances of an individual’s health status. Similarly, the FI, with its extensive list of 32 self-reported measures, can be time-consuming and burdensome to administer in a clinical setting. In contrast, the 3-item HI streamlines the assessment process by focusing on three key, quantitatively measured indicators. This brevity and simplicity make it a more practical tool for routine health assessments, allowing for the identification of individuals with underlying impairments deriving from multiple physiological systems.

The approach used to define risk groups based on HI scores in this study employed a data-driven methodology. By utilizing multiple univariate Cox models and examining the HRs and *p*-values at various HI cut-off values, we were able to identify optimal thresholds for differentiating between Low-Risk, Medium-Risk, and High-Risk individuals. This data-driven approach provides a systematic and objective way to categorize individuals based on their health status and risk levels. By considering both the HRs and *p*-values, we aimed to strike a balance between the strength of association and statistical significance in defining the cut-off values. The resulting risk groups allow for a more refined assessment of individuals’ health profiles, which can aid in targeted interventions and the allocation of resources to those at higher risk. Moreover, the use of HI as a continuous measure provides a more nuanced representation of health compared with traditional categorical measures, potentially capturing a wider range of health variations within the population. Future studies should validate these risk groups and explore their utility in predicting other health outcomes and guiding personalized interventions.

To facilitate the use of, and to encourage other researchers to test the utility of the new 3-item HI, an easy-to-use MATLAB app was also created and made freely available for download. Researchers interested in implementing the HI in their studies can access and utilize the app and code to facilitate data processing and analysis. Overall, the development of the MATLAB app enhances the accessibility and practicality of the new 3-item HI, enabling its broader adoption and further exploration in clinical research settings.

The strengths of our study include the large sample size, long-term follow-up, and comprehensive adjustment for confounding variables. These factors enhance the robustness and generalizability of our findings. Moreover, the availability of multiple health measures allowed for a direct comparison of their predictive abilities. The 3-item HI showed promise as a simpler and more efficient quantitative tool that can be ultimately incorporated into routine clinical assessments.

However, it is important to acknowledge several limitations of this study. First, the findings are based on a specific cohort and may not be fully representative of the general population. This is confounded by the high exclusion rates. The main reason for exclusion from this study (*N* = 3140) was because those participants did not attend one of the dedicated health assessment centers (located in Dublin and Cork, Ireland), and as such the required data were not available for those participants. Previous TILDA work has shown that, among other differences, respondents who did not attend the health assessment center had higher levels of physical disability, were weaker by grip strength, and had slower walking speed [64]. Appendix F provides further insights into the demographics of excluded and included participants. Notably, the excluded group had a higher percentage of deceased participants (24% vs. 10%) and was older (median age: 64 vs. 60 years). Differences in sex and educational attainment were observed, with self-reported diabetes being more prevalent among excluded participants (9% vs. 6%). Variations in factors like cardiovascular diseases and antihypertensive medication also suggest the potential for selection bias. Replication of these results in diverse populations is necessary to establish the external validity of the 3-item HI. Despite our efforts to control for confounding variables, residual confounding or unmeasured factors may influence the observed associations. Also, the normalization process used for the 3-item HI relies on assumptions based on prior literature and may require further validation. Additionally, while our study employed Cox proportional hazards regression to analyze mortality risk, it is important to acknowledge the potential limitation of not exploring parametric models. While the underlying hazard function’s smoothness might suggest a parametric approach, the Cox proportional hazards model’s flexibility allowed us to capture complex and evolving risk dynamics without making strong assumptions about hazard distribution. Future investigations can consider parametric models to further assess the robustness of our findings and ascertain whether the observed associations persist under different modeling assumptions.

Further research is warranted to advance our understanding of the 3-item HI and refine its clinical utility. Future studies should include longitudinal designs to validate its predictive ability across diverse populations and assess its long-term stability. Additionally, exploring the integration of additional variables or modifications to the health measures may enhance their accuracy and clinical relevance.

## 5. Conclusions

The new 3-item HI, which was solely based on objective measures across cardiovascular, neurocognitive and locomotor domains, showed better discrimination than two frailty measures (32-item FI and FP) at predicting 12-year cardio-respiratory mortality in TILDA. This suggests that, subject to future external validation in clinical research settings, this new tool can be more useful than existing frailty measures in the prediction of cardiovascular and respiratory risk.

## Figures and Tables

**Figure 1 diagnostics-13-02801-f001:**
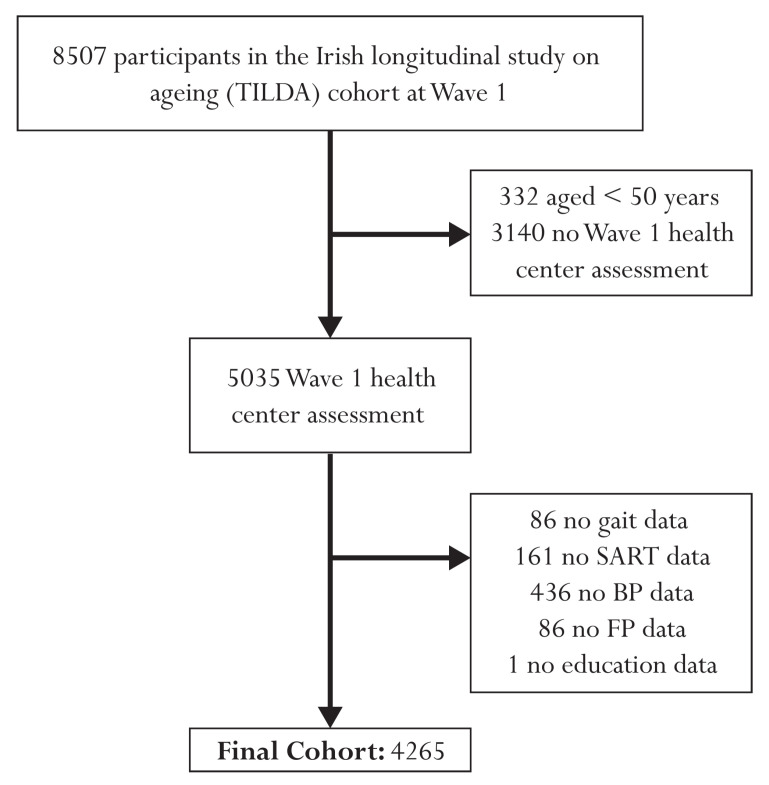
Flow chart describing sample selection and exclusions.

**Figure 2 diagnostics-13-02801-f002:**
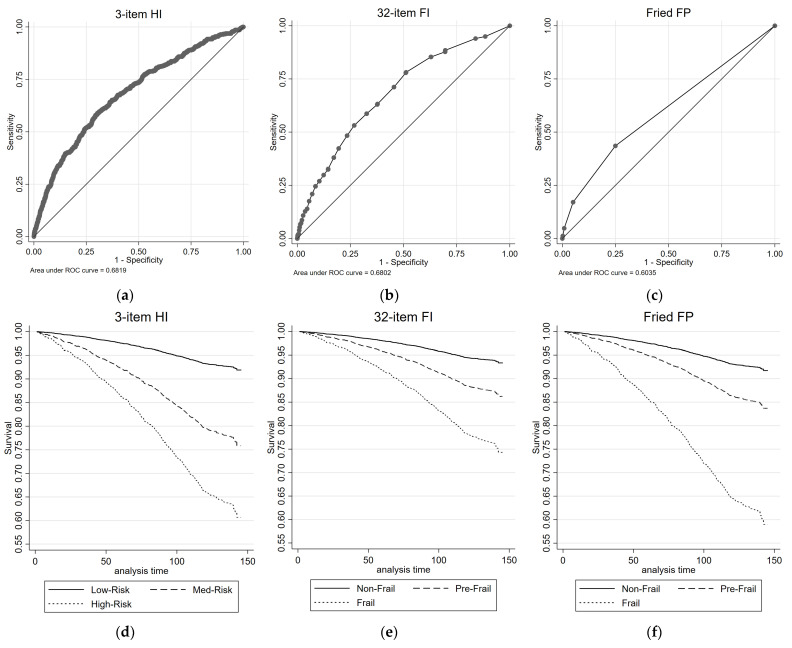
ROC and survival plots for all-cause mortality prediction for the: (**a**,**d**) 3-item HI; (**b**,**e**) 32-item FI; and (**c**,**f**) Fried FP.

**Figure 3 diagnostics-13-02801-f003:**
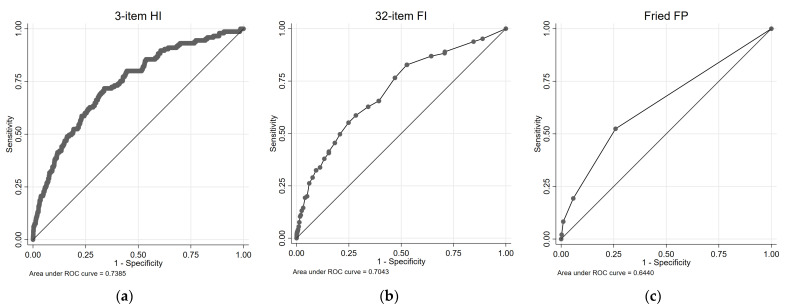
ROC and survival plots for combined cardiovascular/respiratory mortality prediction for the: (**a**,**d**) 3-item HI; (**b**,**e**) 32-item FI; and (**c**,**f**) Fried FP.

**Table 1 diagnostics-13-02801-t001:** Demographic characteristics of the study samples.

	Total	Low-Risk	Medium-Risk	High-Risk
	*N* = 4265	*N* = 3604	*N* = 625	*N* = 36
**Mortality Status** (Deceased) (% (*n*))	10% (416)	7% (265)	22% (138)	36% (13)
**sBP SampEn** (mean (SD))	0.640 (0.178)	0.614 (0.166)	0.782 (0.164)	0.843 (0.184)
**Usual Gait Speed** (cm/s) (median (IQR))	136.7 (25.8)	139.6 (24.0)	118.9 (26.6)	105.1 (32.2)
**No. SART Bad Performances ** (median (IQR))	20 (4.0)	2.0 (3.0)	8.0 (7.0)	19.5 (3.5)
**3-item Health Index Score** (median (IQR))	0.345 (0.116)	0.329 (0.094)	0.499 (0.069)	0.681 (0.055)
**32-item Frailty Index** (% (*n*))				
Non-frail	60% (2553)	64% (2313)	36% (228)	33% (12)
Pre-frail	30% (1281)	29% (1034)	38% (236)	31% (11)
Frail	10% (431)	7% (257)	26% (161)	36% (13)
**Fried Frailty Phenotype** (% (*n*))				
Non-frail	73% (3123)	77% (2772)	55% (341)	28% (10)
Pre-frail	26% (1088)	23% (811)	41% (258)	53% (19)
Frail	1% (54)	1% (21)	4% (26)	19% (7)
**Age** (years) (median (IQR))	60.0 (12.0)	59.0 (12.0)	67.0 (13.0)	73.5 (10.5)
**Sex** (Female) (% (*n*))	54% (2311)	52% (1891)	63% (392)	78% (28)
**Highest education achieved** (% (*n*))				
Primary/none	21% (896)	17% (626)	40% (250)	56% (20)
Secondary	42% (1771)	42% (1522)	38% (238)	31% (11)
Third/higher	37% (1598)	40% (1456)	22% (137)	14% (5)
**BMI** (kg/m^2^) (median (IQR))	28.1 (6.0)	27.8 (5.7)	29.9 (6.2)	29.3 (5.3)
**Self-reported diabetic** (% (*n*))	6% (260)	5% (178)	12% (76)	17% (6)
**No. of Cardiovascular Diseases** (% (*n*))				
None	83% (3523)	84% (3031)	75% (467)	69% (25)
1	13% (557)	12% (440)	17% (109)	22% (8)
2 or more	4% (185)	4% (133)	8% (49)	8% (3)
**Antihypertensive Medications** (Yes) (% (*n*))	33% (1399)	30% (1072)	49% (306)	58% (21)
**Smoker** (% (*n*))				
Never	46% (1975)	47% (1694)	42% (262)	53% (19)
Past	39% (1669)	39% (1405)	40% (252)	33% (12)
Current	15% (621)	14% (505)	18% (111)	14% (5)
**CAGE Alcohol Scale** (% (*n*))				
CAGE < 2	78% (3342)	78% (2820)	79% (496)	72% (26)
CAGE ≥ 2	13% (549)	14% (487)	10% (60)	6% (2)
No response	9% (374)	8% (297)	11% (69)	22% (8)
**Number of ADL impairments** (% (*n*))				
0	94% (4011)	95% (2922)	91% (1060)	81% (29)
1	4% (188)	4% (110)	6% (74)	11% (4)
2 or more	2% (66)	1% (31)	3% (32)	8% (3)
**Number of IADL impairments** (% (*n*))				
0	96% (4115)	98% (2991)	94% (1094)	83% (30)
1	3% (107)	2% (52)	4% (51)	11% (4)
2 or more	1% (43)	1% (20)	2% (21)	6% (2)

Abbreviations: interquartile range (IQR); systolic blood pressure (sBP); sample entropy (SampEn); sustained attention reaction time (SART); body mass index (BMI); activities of daily living (ADLs); instrumental activities of daily living (IADLs).

**Table 2 diagnostics-13-02801-t002:** Results from Cox proportional regression and ROC analyses for all-cause mortality prediction.

Mortality Prediction (All-Cause, 416 Deaths)						
		Univariate		Multivariate		
Measure	*N*	HR (95% CIs)	*p*-Value	HR (95% CIs)	*p*-Value	AUC
3-item Health Index						0.68
Low-Risk (ref)	3604	1		1		
Medium-Risk	625	3.27 (2.66, 4.01)	**<0.001**	1.75 (1.40, 2.19)	**<0.001**	
High-Risk	36	5.91 (3.38, 10.34)	**<0.001**	2.06 (1.16, 3.65)	**0.013**	
32-item Frailty Index						0.68
Non-Frail (ref)	2553	1		1		
Pre-Frail	1281	2.15 (1.72, 2.69)	**<0.001**	1.19 (0.92, 1.54)	**0.182**	
Frail	431	4.31 (3.36, 5.53)	**<0.001**	1.68 (1.22, 2.32)	**0.001**	
Fried Frailty Phenotype						0.60
Non-Frail (ref)	3123	1		1		
Pre-Frail	1088	2.06 (1.69, 2.52)	**<0.001**	1.34 (1.08, 1.65)	**0.007**	
Frail	54	6.11 (3.84, 9.71)	**<0.001**	2.26 (1.42, 3.60)	**0.001**	

**Table 3 diagnostics-13-02801-t003:** Results from Cox proportional regression and ROC analyses for combined cardiovascular/respiratory mortality prediction.

Mortality Prediction (Cardiovascular and Respiratory, 145 Deaths)						
		Univariate		Multivariate		
Measure	*N*	HR (95% CIs)	*p*-Value	HR (95% CIs)	*p*-Value	AUC
3-item Health Index						0.74
Low-Risk (ref)	3604	1		1		
Medium-Risk	625	4.32 (3.06, 6.09)	**<0.001**	2.17 (1.47, 3.19)	**<0.001**	
High-Risk	36	15.10 (7.79, 29.28)	**<0.001**	5.61 (2.84, 11.05)	**<0.001**	
32-item Frailty Index						0.70
Non-Frail (ref)	2553	1		1		
Pre-Frail	1281	1.99 (1.34, 2.96)	**0.001**	0.91 (0.58, 1.43)	**0.690**	
Frail	431	6.12 (4.11, 9.11)	**<0.001**	1.73 (1.02, 2.92)	**0.041**	
Fried Frailty Phenotype						0.64
Non-Frail (ref)	3123	1		1		
Pre-Frail	1088	2.79 (1.99, 3.91)	**<0.001**	1.72 (1.21, 2.46)	**0.003**	
Frail	54	12.53 (6.76, 23.26)	**<0.001**	3.31 (1.69, 6.49)	**0.001**	

## Data Availability

The datasets generated during and/or analyzed during the current study are not publicly available due to data protection regulations but are accessible at TILDA on reasonable request. The procedures to gain access to TILDA data are specified at https://tilda.tcd.ie/data/accessing-data/ (accessed on 22 May 2023).

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
