# Peer review of "Evaluation of a 3-Item Health Index in Predicting Mortality Risk: A 12-Year Follow-Up Study"

_diagnostics, 2023, doi:10.3390/diagnostics13172801_

Round 1

Reviewer 1 Report

l138: This is a mathematical overview. There's little difference between B^m_i and A^m_i; as stated, B^m_i=A^{m+1}_i. Is this correct?
Can you state in words what equations (1) and (2), and then (3) represent? Readers can be assumed to understand mean and standard deviation, and entropy should be related to these concepts.

Equation (5) is unnecessarily complicated. Note that (x-1) time (-1) is 1-x, so use (UGS_{max}-UGS)/UGC_{range}. I recommend changing the MATLAB also, as simpler forms are typically more stable numerically.

l217 the 'mean' or 'arithmetic mean', not 'average'.

Perhaps the mathematical formulae could be moved to an appendix.

2.6. 3-item HI risk groups
This is an approach which is likely to result in over-fitting. It is not well specified: 'Multiple univariate Cox models' might be using various possible outcomes. Was the outcome time to death from any cause?
Figure 1A shows how arbitrary the choices are. This figure must include the confidence intervals for the hazard ratio.

l 283: why throw away information by categorising BMI? Is the main reason to match with the WHO guidelines? A simple cubic could be fitted.

2.10. Statistical Analysis
l289 mean and sd are not always best for continuous variables.
l299 Why was Cox PH used? Was there no parametric model which fitted well?
In this example, the underlying hazard function is of interest, and is likely to be smooth, so the loss of information due to a semi-parametric approach is not justified.

l290 you also have polytomous variables: suggest using 'categorical variables'.
l291-293 Spearman's rank correlation is loved by social scientists, but it is not a sensible measure, as it is not easy to interpret (though social scientists tend to interpret it as if it were a Pearson correlation). In terms of effective description, it is better to give sensible summary statistics for continuous variables for each of the categories.

l308-l311. What is the point of adjusting? You must give an explicit reason.
Surely you are aiming to show that a simply measure, HI, can replace the list "ge, sex, education, BMI, antihypertensive medication use, diabetes status, number of cardiovascular conditions, smoking status, and alcohol consumption," which includes variables which are often not recorded or are not reliable.

Figure 1. Add the percentages excluded, as this is essential to discussion of generalisability and bias. Almost half the study population was excluded. You ought to give summary demographic and survival statistics for men and women included (4265) and excluded (>50, 3910), to inform this discussion.

l352-l355. In general, do not give p-values before actual results. Further, the p-value in this instance is not valid, as it is a function of your procedure to define the cut points.
It does *not* 'underscores the association' l356.
Also, with a very small group in high-risk (36 out of 4265, <1%) , there will be limited information.

Table 1. columns of numbers must be right-aligned.
'No. SART Bad Performances' is clearly skew. It should be given as percentage of zeroes, and then a summary of non-zero values, either Lq, median, Uq, or a sensible categorisation.
FI and FP: is the categorisation standard? If so, state cut-points, and reference. If not, give sensible summary statistics, such as Lq, median, Uq.
I see that HI high risk rates are constant across HI groups.

l379-382: the p-value in this instance is not valid, as it is a function of your procedure to define the cut points. HI and FI are not distinguishable. HI does not show better performance.

l430-433: HI and FI are not distinguishable. HI does not show better performance.

Figure 2, d, e, f should have the same vertical axes.

The discussion must include more informed comments on the biases likely, in the light of sensitivity analysis.

Any revision must be accompanied by the STARD checklist
(https://www.equator-network.org/reporting-guidelines/stard/)

The article should be more concisely written.

Author Response

Comments and Suggestions for Authors

We sincerely appreciate the reviewer’s thorough and helpful suggestions, which have resulted in a much improved manuscript.

l138: This is a mathematical overview. There's little difference between B^m_i and A^m_i; as stated, B^m_i=A^{m+1}_i. Is this correct?
Can you state in words what equations (1) and (2), and then (3) represent? Readers can be assumed to understand mean and standard deviation, and entropy should be related to these concepts.

Equation (1) calculates B^m(r), which represents the average count of similar template vectors of length m within the time series. These vectors are counted and averaged across all possible starting points in the time series. Similarly, equation (2) calculates A^m(r), the average count of similar template vectors of length m + 1.

We agree with the reviewer's observation that B^m_i is very similar to A^m_i. Both B^m_i and A^m_i represent the count of similar template vectors within their respective lengths, but the lengths differ by one unit (as the Reviewer also observed). This similarity arises from the mathematical calculation and the nature of the template matching process. In practical terms, this means that we are comparing the similarity of slightly longer sequences in A^m(r) compared to B^m(r).

One could perhaps use a more intuitive notation, e.g. A^m(r) and A^{m+1}(r), however we decided, for consistency, to follow the ‘A/B’ convention that was first used in the seminal Sample Entropy paper [1], and has subsequently been used many times in other works (including our own), e.g.: [2-4].

Moving to equation (3), SampEn(m, r, N) quantifies the relative unpredictability or randomness in the time series. It does this by comparing the counts of similar template vectors for lengths m and m + 1. The division of A^m(r) by B^m(r) provides a ratio that reflects how many patterns of length m + 1 are found vs. patterns of length m. The negative natural logarithm of this ratio results in the SampEn value.

We have added a short paragraph after the mathematical description giving a description of the process in words:

Lines 510-515: ‘In simpler terms, SampEn (as calculated in Equation 3) captures the likelihood of finding similar patterns in the data, considering different lengths of sequences. When the ratio of longer patterns (A^m (r): Equation 2) to shorter patterns (B^m (r): Equation 1) is smaller, it suggests greater complexity and unpredictability in the data, leading to a higher SampEn value. Conversely, a larger ratio indicates more regularity and results in a lower SampEn value.’

We have also added ‘… * the standard deviation (SD) of the timeseries)’ after the introduction of the threshold r, in order to better explain how the thresholding works in relation to the SD of the data.

Equation (5) is unnecessarily complicated. Note that (x-1) time (-1) is 1-x, so use (UGS_{max}-UGS)/UGC_[5]. I recommend changing the MATLAB also, as simpler forms are typically more stable numerically.

Thank you for this input, we agree. The equation has been simplified down as suggested (I also doubled checked and changing this in the code resulted in the same output values).

l217 the 'mean' or 'arithmetic mean', not 'average'.

Thank you for another good point, it has been changed to ‘mean’.

Perhaps the mathematical formulae could be moved to an appendix.

We have moved the formulae for calculating the HI to an Appendix.

2.6. 3-item HI risk groups
This is an approach which is likely to result in over-fitting. It is not well specified: 'Multiple univariate Cox models' might be using various possible outcomes. Was the outcome time to death from any cause?

The outcome was all-cause mortality; this information has now been added to the manuscript, both at the instance described above, and also in the title to Figure 1B.

Figure 1A shows how arbitrary the choices are. This figure must include the confidence intervals for the hazard ratio.

Thank you for this input, 95% CIs have been added to HR values on plots.

We appreciate the reviewer's thoughtful observation regarding the choice of cut-off values in Figure B1 (previously A1). We acknowledge that such choices can indeed be seen as arbitrary to some extent. Our selection of cut-offs was primarily motivated by the intention to enable comparison with widely used frailty measures, typically reported in stratified groups, and to enhance the reader's intuitive understanding of the results. Figure B1 (previously A1) also serves as a sensitivity analysis, demonstrating that a range of cut-off values, both higher and lower, yield similar statistical outcomes in the fully controlled models. This robustness in the face of varying cut-offs lends support to the potential utility and broader applicability of the HI measure.

We have now included the following paragraph in the Discussion section highlighting this important point:

Lines 1880-1894: ‘Figure B1 offers valuable insights into the robustness of the HI measure's performance across a range of cut-off values. Notably, the plot demonstrates that various cut-offs produce similar statistical results in the fully controlled models. This finding enhances confidence in the HI measure's effectiveness and highlights its potential as a versatile tool for assessing mortality risk. The consistency of outcomes across a spectrum of cut-offs underscores the measure's stability and lends further support to its potential applicability in different clinical contexts. While this derivation of the optimal risk cut-off groups contributes to the understanding of the HI measure's performance, it also emphasizes the need for continued research in refining the optimal cut-off value, potentially incorporating domain-specific considerations. Further exploration is warranted to ascertain whether specific population characteristics or health conditions could influence the choice of cut-off and enhance the measure's predictive precision. Additionally, investigating the longitudinal stability of the chosen cut-off values and their generalizability to diverse populations would provide a comprehensive perspective on the HI measure's utility, as has been previously performed with frailty measures [62, 63].’

l 283: why throw away information by categorising BMI? Is the main reason to match with the WHO guidelines? A simple cubic could be fitted.

While our study primarily focuses on the predictive capacity of the HI variable, we utilized BMI categorization to establish the variable's robustness within a comprehensive model that accounts for multiple risk factors. We appreciate your suggestion of exploring a simple cubic fitting method to address non-linearity, and we recognize the potential merit of this alternative approach.

It is noteworthy that many prior publications also employ BMI categorization, which aligns well with our decision. For example, a 2016 review/meta-analysis of obesity in 200,777 participants from 15 international studies reported that ‘10 studies used International Obesity Task Force/Cole 2000/World Health Organization reference populations (36); others used the Centers for Disease Control and Prevention 2000 reference growth charts (37) or the UK 90 reference (38). All assessments of adult weight status used the BMI cut-offs for overweight and obesity of 25 and 30 kg m-2’ [6]. This observation underscores the broader relevance of our categorization method. However, we are mindful of your suggestion and will certainly take it into account for future investigations that may centre on BMI or obesity as the primary focus.

Importantly, we acknowledge the potential differences in robustness between the categorization and cubic fit approaches. It is worth noting that our study is directed toward a medical/clinical audience who are accustomed to the categorization guidelines provided by the World Health Organization (WHO). As such, our decision to employ BMI categorization aligns with the expectations and interpretative frameworks commonly used in this field.

2.10. Statistical Analysis
l289 mean and sd are not always best for continuous variables.

This is a very good point, and we thank the reviewer for their input. We have now investigated the distributions of all continuous variables reported, by visual examinations of the histograms and tested using the skewness kurtosis test for normality. Those normally distributed are reported as mean/SD, and those non-normally distributed as median/interquartile range (IQR). These have been updated in both Table 1 and the new Table E1.

Lines 693-699: ‘Visual examination was conducted on the distributions of continuous variables using histograms and distributional diagnostic plots and tested using the skewness kurtosis test for normality. Summary statistics of the overall cohort and HI groups ('Low-Risk', 'Medium-Risk', 'High-Risk') were calculated as mean and SD for continuous normally dis-tributed variables, while non-normally distributed continuous variables were summarized using the median and interquartile range (IQR). Proportions were presented as both counts and percentages.’

l299 Why was Cox PH used? Was there no parametric model which fitted well?
In this example, the underlying hazard function is of interest, and is likely to be smooth, so the loss of information due to a semi-parametric approach is not justified.

The choice of Cox PH regression over a parametric model was based on several factors, including the nature of the data and the goals of our analysis. Our response aims to clarify these aspects and provide insight into our rationale for selecting Cox PH regression.

Nature of the Data: Our study involves a large cohort of older adults with a 12-year follow-up period, during which the occurrence of mortality events was being monitored. The hazard function, representing the instantaneous rate of death at a given time, can be complex and nonlinear over time, especially in a heterogeneous population with varying risk factors. The Cox PH model is well-suited for handling time-to-event data when the shape of the hazard function is not precisely known or when it may change over time.

Model Flexibility: While parametric models assume a specific functional form for the hazard function (e.g., exponential, Weibull), the Cox PH model remains flexible by allowing the hazard to vary non-parametrically with time. This flexibility is particularly advantageous when dealing with situations where the true underlying hazard shape is uncertain or when the data do not conform neatly to a parametric distribution.

Interpretability and Generality: Cox PH regression provides interpretable hazard ratios, which are essential for quantifying the effect of covariates on survival. Additionally, the Cox PH model is general and does not require assumptions about the distribution of survival times, making it a robust choice for a wide range of survival analysis scenarios. The main intended audience for this work (i.e. health sciences or clinical) will also be very familiar with interpreting results from this approach.

While parametric models could be considered as an alternative approach, we believe that the Cox PH model was appropriate for our study objectives, given the potential complexity of the hazard function and the potential for unobserved heterogeneity in our population. Implementing a parametric model might introduce assumptions about the shape of the hazard function that may not accurately reflect the underlying dynamics of mortality risk in our dataset. We do acknowledge the potential advantages of parametric models in capturing certain types of hazard functions and will carefully consider this approach in future research. We also now acknowledge this as a potential limitation of the study in the Discussion section with the inclusion of the following text:

Lines 1964-1972: ‘Additionally, while our study employed Cox proportional hazards regression to analyze mortality risk, it is important to acknowledge the potential limitation of not exploring parametric models. While the underlying hazard function's smoothness might suggest a parametric approach, the Cox proportional hazards model's flexibility allowed us to capture complex and evolving risk dynamics without making strong assumptions about hazard distribution. Future investigations could consider parametric models to further assess the robustness of our findings and ascertain whether the observed associations persist under different modeling assumptions.’

l290 you also have polytomous variables: suggest using 'categorical variables'.

Thank you, this has been amended.

l291-293 Spearman's rank correlation is loved by social scientists, but it is not a sensible measure, as it is not easy to interpret (though social scientists tend to interpret it as if it were a Pearson correlation). In terms of effective description, it is better to give sensible summary statistics for continuous variables for each of the categories.

Thank you for your input. In order to remove any confusion, and avoid misinterpretation of Table 1, we have now removed the Spearman's rank correlation results, and instead report sensible summary statistic, chosen depending on the type and distribution of a particular variable.

l308-l311. What is the point of adjusting? You must give an explicit reason.
Surely you are aiming to show that a simply measure, HI, can replace the list "ge, sex, education, BMI, antihypertensive medication use, diabetes status, number of cardiovascular conditions, smoking status, and alcohol consumption," which includes variables which are often not recorded or are not reliable.

Thank you for raising this important question. The primary reason for adjusting in our analysis is to account for potential confounding effects. The adjustment allows us to assess whether the HI retains its predictive ability even in the presence of other factors that clinicians know could impact mortality risk. We agree that the goal is to show the utility of the HI as a replacement for a more complex list of variables. However, by conducting both unadjusted and fully adjusted models, we ensure a comprehensive evaluation of the HI's predictive performance and its potential to replace or enhance existing measures. We have now clarified this in the manuscript with the inclusion of the following text:

Lines 767-772: ‘In order to assess the independent predictive power of the 3-item HI, we used two models: (i) an unadjusted model and (ii) a fully adjusted model. The adjustment aimed to account for potential confounding effects that may arise from covariates commonly associated, clinically and / or epidemiologically, with mortality risk. These covariates included age, sex, education, BMI, antihypertensive medication use, diabetes, number of cardiovascular conditions, smoking status, and alcohol consumption.’

Figure 1. Add the percentages excluded, as this is essential to discussion of generalisability and bias. Almost half the study population was excluded. You ought to give summary demographic and survival statistics for men and women included (4265) and excluded (>50, 3910), to inform this discussion.

This is a very good point. We have now included comparative demographics for the full cohort, and the excluded and included cohorts as an Appendix to the paper. We have also added the following to the Discussion section reporting on this limitation to the study:

Lines 1939-1954: ‘However, it is important to acknowledge several limitations of this study. First, the findings are based on a specific cohort and may not be fully representative of the general population. This is confounded by the high exclusion rates. The main reason for exclusion from this study (N = 3140) was because those participants did not attend one of the dedicated health assessment centers (located in Dublin and Cork, Ireland), and as such the required data was not available for those participants. Previous TILDA work has shown that, among other differences, respondents who did not attend the health assessment center had higher levels of physical disability, were weaker by grip strength, and had slower walking speed [64]. Appendix E provides further insights into the demographics of excluded and included participants. Notably, the excluded group had a higher percentage of deceased participants (24% vs. 10%) and was older (median age: 64 vs. 60 years). Differences in sex and educational attainment were observed, with self-reported diabetes being more prevalent among excluded participants (9% vs. 6%). Variations in factors like cardiovascular diseases and antihypertensive medication also suggest the potential for selection bias. Replication of these results in diverse populations is necessary to establish the external validity of the 3-item HI’

l352-l355. In general, do not give p-values before actual results. Further, the p-value in this instance is not valid, as it is a function of your procedure to define the cut points.
It does *not* 'underscores the association' l356.

The reviewer makes a good point; the p-value was removed, as well as the referenced sentence.

Also, with a very small group in high-risk (36 out of 4265, <1%) , there will be limited information.

This is also a good point. However, as discussed above, Figure B1 clearly demonstrates the robustness of the cut-off point (acting as a sensitivity analysis in this instance), and as such we believe our choice of smaller groups to optimise the hyperparameter tuning approach for the choice of cut-offs is justified in this instance.

Table 1. columns of numbers must be right-aligned.

Amended.

'No. SART Bad Performances' is clearly skew. It should be given as percentage of zeroes, and then a summary of non-zero values, either Lq, median, Uq, or a sensible categorisation.

Again, thank you for your input on this point. As stated above, non-normally distributed continuous variables (including 'No. SART Bad Performances') are now reported as medians and IQRs.

FI and FP: is the categorisation standard? If so, state cut-points, and reference. If not, give sensible summary statistics, such as Lq, median, Uq. I see that HI high risk rates are constant across HI groups.

Both the FI and FP categorisations were based on established cut-offs previously reported and used extensively in the literature. To clarify this the following has been added to 2.8. Frailty measures:

Lines 664-667: ‘As per previous literature [3, 22, 45, 46], ‘frail’ was defined as having 3 or more criteria (from low grip strength, low physical activity, low gait speed, unintentional weight loss, or self-reported exhaustion), ‘pre-frail’ as having 1 or 2 criteria, and ‘non-frail’ as having none.’

Lines 670-671: In line with previous literature [52], the following cut-offs were applied for the definition of the three FI states: FI < 0.10, ‘non-frail’; 0.10 ≤ FI < 0.25, ‘pre-frail’; and FI ≥ 0.25, ‘frail’.’

l379-382: the p-value in this instance is not valid, as it is a function of your procedure to define the cut points.

Thank you, p-values have been removed.

l430-433: HI and FI are not distinguishable. HI does not show better performance.

That comment in the manuscript is specifically in reference to cardiovascular and respiratory death prediction, not all-cause mortality where indeed HI and FI performed similarly. Clarification has been made in the manuscript.

Figure 2, d, e, f should have the same vertical axes.

Thank you for that observation, we have now amended this in the manuscript. We have also added the leading zeros into the survival plots y-axis labels (including Figure 3).

The discussion must include more informed comments on the biases likely, in the light of sensitivity analysis.

The Discussion section now includes discussions related to potential bias originating from study exclusions and sensitivity analysis related to the choice of cut-off for the higher risk group (please see above).

Any revision must be accompanied by the STARD checklist
(https://www.equator-network.org/reporting-guidelines/stard/)

Comments on the Quality of English Language

The article should be more concisely written.

We hope that the revisions have resulted in more precise and concise reporting, while retaining the full information necessary for readers to appreciate and understand the complete research process behind the new HI measure.

Submission Date

19 July 2023

Date of this review

09 Aug 2023 16:51:49

[1]          J. S. Richman and J. R. Moorman, "Physiological time-series analysis using approximate entropy and sample entropy," (in eng), Am J Physiol Heart Circ Physiol, vol. 278, no. 6, pp. H2039-49, Jun 2000, doi: 10.1152/ajpheart.2000.278.6.H2039.

[2]          R. Goya-Esteban, J. P. M. d. Sa, J. L. Rojo-Alvarez, and O. Barquero-Perez, "Characterization of Heart Rate Variability loss with aging and heart failure using Sample Entropy," in 2008 Computers in Cardiology, 14-17 Sept. 2008 2008, pp. 41-44, doi: 10.1109/CIC.2008.4748972.

[3]          A. Delgado-Bonal and A. Marshak, "Approximate Entropy and Sample Entropy: A Comprehensive Tutorial," (in eng), Entropy (Basel), vol. 21, no. 6, May 28 2019, doi: 10.3390/e21060541.

[4]          D. Bajić and N. Japundžić-Žigon, "On Quantization Errors in Approximate and Sample Entropy," Entropy, vol. 24, no. 1, p. 73, 2022. [Online]. Available: https://www.mdpi.com/1099-4300/24/1/73.

[5]          A. Afshin et al., "Health Effects of Overweight and Obesity in 195 Countries over 25 Years," (in eng), N Engl J Med, vol. 377, no. 1, pp. 13-27, Jul 6 2017, doi: 10.1056/NEJMoa1614362.

[6]          M. Simmonds, A. Llewellyn, C. G. Owen, and N. Woolacott, "Predicting adult obesity from childhood obesity: a systematic review and meta-analysis," Obesity Reviews, vol. 17, no. 2, pp. 95-107, 2016, doi: https://doi.org/10.1111/obr.12334.

Please find below STARD checklist for this manuscript.

Section & Topic

No

Item

Reported on page #

TITLE OR ABSTRACT

1

Identification as a study of diagnostic accuracy using at least one measure of accuracy

(such as sensitivity, specificity, predictive values, or AUC)

7

ABSTRACT

2

Structured summary of study design, methods, results, and conclusions
(for specific guidance, see STARD for Abstracts)

1

INTRODUCTION

3

Scientific and clinical background, including the intended use and clinical role of the index test

1-3

4

Study objectives and hypotheses

3

METHODS

Study design

5

Whether data collection was planned before the index test and reference standard
were performed (prospective study) or after (retrospective study)

3

Participants

6

Eligibility criteria

3

7

On what basis potentially eligible participants were identified
(such as symptoms, results from previous tests, inclusion in registry)

3

8

Where and when potentially eligible participants were identified (setting, location and dates)

3

9

Whether participants formed a consecutive, random or convenience series

3

Test methods

10a

Index test, in sufficient detail to allow replication

3-7, 16-21

10b

Reference standard, in sufficient detail to allow replication

5

11

Rationale for choosing the reference standard (if alternatives exist)

N/A

12a

Definition of and rationale for test positivity cut-offs or result categories
of the index test, distinguishing pre-specified from exploratory

5-6

12b

Definition of and rationale for test positivity cut-offs or result categories
of the reference standard, distinguishing pre-specified from exploratory

5

13a

Whether clinical information and reference standard results were available
to the performers/readers of the index test

N/A

13b

Whether clinical information and index test results were available
to the assessors of the reference standard

N/A

Analysis

14

Methods for estimating or comparing measures of diagnostic accuracy

6-7

15

How indeterminate index test or reference standard results were handled

7

16

How missing data on the index test and reference standard were handled

7-8, 21-22

17

Any analyses of variability in diagnostic accuracy, distinguishing pre-specified from exploratory

6-7

18

Intended sample size and how it was determined

3, 7-8

RESULTS

Participants

19

Flow of participants, using a diagram

8

20

Baseline demographic and clinical characteristics of participants

8-9, 21-22

21a

Distribution of severity of disease in those with the target condition

8-9

21b

Distribution of alternative diagnoses in those without the target condition

8-9

22

Time interval and any clinical interventions between index test and reference standard

3, 5

Test results

23

Cross tabulation of the index test results (or their distribution)
by the results of the reference standard

8-9

24

Estimates of diagnostic accuracy and their precision (such as 95% confidence intervals)

9-13

25

Any adverse events from performing the index test or the reference standard

N/A

DISCUSSION

26

Study limitations, including sources of potential bias, statistical uncertainty, and generalisability

15

27

Implications for practice, including the intended use and clinical role of the index test

14-15

OTHER INFORMATION

28

Registration number and name of registry

16

29

Where the full study protocol can be accessed

16

30

Sources of funding and other support; role of funders

16

Reviewer 2 Report

Evaluation of a 3-Item Health Index in Predicting Mortality Risk: a 12-year Follow Up Study

Below are notes to improve the manuscript

The abstract is well written but can be improved. Which tool was used to analyze the collected data.

The introduction section needs to be improved to include the main research problems and research aim.

No research questions were examined. Do provide research questions in the introduction section.

A section on related works needs to be well written. Include prior studies on Mortality Risk Prediction

The methodology looks well written. But provide justification for methodology employed.

What sampling was employed in the study.

Improve the results section and related back to the literature. What are the main discoveries from the study.

The discussion section needs to map back to the literature. This is not well presented. Possibly you can improve this section.

Include implication for theory and implication for practice related Mortality Risk Prediction.

The conclusion should include the significance of the study.

Also, the limitations and future works needs to be described in detail

Good luck

Please do proofread the manuscript

Author Response

Below are notes to improve the manuscript

We thank the reviewer for taking the time to review our work, and we appreciate the constructive comments/input.

The abstract is well written but can be improved. Which tool was used to analyze the collected data.

Thank you for your comment, we have now added details with regards the software used, with the inclusion of the following text:

Lines 19-20: ‘All data processing was performed using MATLAB and statistical analysis using STATA 15.1.’

The introduction section needs to be improved to include the main research problems and research aim.

The Introduction section outlines the aim of the study, as well as the issues we aimed to address with regards current measures, in the second paragraph:

Lines 111-118: ‘In this study, we developed and tested a new health index (HI) that addresses some of the limitations of existing frailty measures by incorporating three objective items: resting-state systolic blood pressure (sBP) sample entropy (SampEn), sustained attention re-action time (SART) performance, and usual gait speed (UGS). These objective and quantitative measures were chosen based on their sensitivity to changes in cardiovascular, neurocognitive, and locomotor status, respectively, and their potential to provide a more comprehensive assessment of an individual's mortality risk based on three combined critical, core physiological systems.’

We then go on to outline and discuss the three measures we chose in the context of the literature. Finally in the Introduction section we recapitulate the aim of the study in the form of the newly added formalised research questions.:

Lines 374-380: ‘Specifically, this study sought to address the following research questions:

  1. How does the newly developed 3-item HI, integrating objective physiological markers such as resting-state sBP SampEn, SART performance, and UGS, compare to frailty measures such as the 32-item FI and FP in predicting all-cause mortality risk?
  2. To what extent does the 3-item HI compare to the above frailty measures in identifying individuals at heightened risk of specific-cause mortality (e.g. cardio-respiratory)?’

No research questions were examined. Do provide research questions in the introduction section.

Thank you for your comment. Research questions have now been added to the Introduction section (please see above).

A section on related works needs to be well written. Include prior studies on Mortality Risk Prediction

We have now included the following short paragraph in the Introduction, which introduces previous studies on mortality risk prediction used frailty and other indices:

Lines 150-158: ‘The 3-item HI is a continuous measure ranging from 0 (lowest risk) to 1 (highest risk) and demonstrates a relatively normal distribution. Unlike many existing frailty measures, which have limitations related to the count-based, largely self-reported approach and predefined criteria [3, 4, 31], the HI relies on objective and continuous health measures and a data-driven quantification of their combined dysregulation. Notably, other re-searchers previously proposed mortality-specific indices, employing models based on 10 to 14 predictor variables, which significantly predicted mortality [32-34]. Yet, these models continue to rely on self-reported data and we therefore hypothesized that the new 3-item HI would predict mortality more accurately than existing measures.’

The methodology looks well written. But provide justification for methodology employed.

Thank you. We have now included the following justification for the choice of methodologies for the 3-item HI derivation:

Lines 570-575: ‘The chosen methodology was underpinned by deliberate considerations to address the research objectives effectively while leveraging the large TILDA dataset to derive HI scores normalized to the large cohort. Notably, the formulation of the 3-item HI aimed to serve as a concise yet comprehensive measure of health, aggregating key objective variables – sBP SampEn, NBP in the SART task, and UGS – that independently were previously associated with negative health outcomes, including mortality risk [8-12, 22, 23, 26-30].’

What sampling was employed in the study.

Sampling information for the study has now been added to the Methods section, with the inclusion of the following text:

Lines 391-396: ‘The study's sample was drawn from the Irish Geodirectory, a comprehensive and current database containing all residential addresses across the Republic of Ireland, compiled by 'An Post' (the Irish Postal Service) and Ordnance Survey Ireland. The initial selection of addresses for the sample was carried out using the RANSAM sampling procedure [33], a multi-stage probability sampling method developed by the Economic and Social Research Institute [32].’

Improve the results section and related back to the literature. What are the main discoveries from the study.

Thank you for this comment. We have now improved the Results section, as well as stating the main results of this study.

The discussion section needs to map back to the literature. This is not well presented. Possibly you can improve this section.

Thank you for your input here. We have expanded the discussion of our results in the context of the literature in the Discussion section as follows:

Lines 1555-1578: ‘In terms of all-cause mortality, the discriminatory abilities of the 3-item HI, FI, and FP were comparable, with similar AUC values. Comparable HR values were also observed between respective groups for all-cause mortality. These findings are consistent with previous studies, including a 2022 meta-analysis of 58 previous studies (pooled N = 1,852,951, pooled number of deaths = 145,276) by Peng et al., which reported pooled HRs for specific-cause and all-cause mortality risk associated with frailty status (assessed using FI, FP, or FRAIL scale). For example, the meta-analysis reported pooled HRs of 1.42 (36 studies) for pre-frail individuals and 2.40 (48 studies) for frail individuals for the prediction of all-cause mortality, compared to the non-frail [57]. It is worth noting that other previously reported mortality-specific predictor models outperform both the 3-item HI and the frailty measures concerning all-cause mortality. Notably, reported AUCs of 0.774 (TILDA; Ireland) [32], 0.859 (UK; English Longitudinal Study of Ageing (ELSA)) [33], and 0.82 (USA; Health and Retirement Study (HRS)) [34] have been achieved by these indices. However, it should be acknowledged that these indices rely on 10 to 14 self-reported variables, which were specifically derived from a wide pool of self-report variables (41 to 67) in two of the studies to optimize mortality prediction [32, 34]. Consequently, direct comparisons to the present study are challenging, as our aim was to develop a more objective, data-driven measure. Of the 3 measures that make up the HI, 2 (sBP SampEn [8] and gait speed [59]) have previously been associated with increased risk of mortality; however, since the reported results are ‘per unit’ of SampEn or gait speed, meaningful direct comparison of the individual performance of these components of the HI with the full 3-item HI is not possible. Although there is a lack of previous studies investigating the predictive associations between SART performance and mortality, it is known that impaired neurocognitive performance is associated with higher mortality risk [60, 61].

Notably, when examining the prediction of combined cardio-respiratory mortality, the new 3-item HI demonstrated superior performance compared to the existing frailty measures. The AUC for the 3-item HI was 0.74, while for the FI and FP, the AUC values were 0.70 and 0.64, respectively. Furthermore, the HRs for the 3-item HI were consistently higher than those for the frailty measures. The HR for Medium-Risk individuals assessed using the 3-item HI was 2.17 in fully adjusted models, and for High-Risk individuals, it was 5.61. In contrast, the HR for pre-frail individuals assessed using the FI was 0.91, and for frail individuals, it was 1.73. Similarly, the HR for pre-frail individuals assessed using the FP was 1.72, and for frail individuals, it was 3.31. The HRs associated with the 3-item HI in the current study were also notably higher than those reported by Peng et al. for both cardiovascular (pre-frail: pooled HR = 1.63 (12 studies); frail: pooled HR = 2.64 (13 studies)) and respiratory (pre-frail: pooled HR = 2.16 (4 studies); frail: pooled HR = 4.91 (5 studies)) mortality risk [57].’

Include implication for theory and implication for practice related Mortality Risk Prediction.

We discuss the implications for health assessment theory and clinical practice in the following text from the Discussion section:

Lines 1895-1902 / 1929-1931: ‘The advantages of the 3-item HI may extend beyond its predictive capabilities. The FP relies on count-based measures, which may not fully capture the nuances of an individual's health status. Similarly, the FI, with its extensive list of 32 self-reported measures, can be time-consuming and burdensome to administer in a clinical setting. In contrast, the 3-item HI streamlines the assessment process by focusing on three key, quantitatively measured indicators. This brevity and simplicity make it a more practical tool for routine health assessments, allowing for the identification of individuals with underlying impairments deriving from multiple physiological systems.’

The 3-item HI showed promise as a simpler and more efficient quantitative tool that could be ultimately incorporated into routine clinical assessments.’

The conclusion should include the significance of the study.

Thank you, the following has been added to the Conclusion:

Lines 1965-1970: ‘The new 3-item HI, which was solely based on objective measures across cardiovascular, neurocognitive and locomotor domains, showed better discrimination than two frailty measures (32-item FI and FP) at predicting 12-year cardio-respiratory mortality in TILDA. This suggests that, subject to future external validation in clinical research settings, this new tool could be more useful than existing frailty measures in the prediction of cardio-vascular and respiratory risk.’

Also, the limitations and future works needs to be described in detail

The limitations of the study, and potential for future work section of the Discussion have now been expanded in the manuscript, as follows:

Lines 1938-1964: ‘However, it is important to acknowledge several limitations of this study. First, the findings are based on a specific cohort and may not be fully representative of the general population. This is confounded by the high exclusion rates. The main reason for exclusion from this study (N = 3140) was because those participants did not attend one of the dedicated health assessment centers (located in Dublin and Cork, Ireland), and as such the required data was not available for those participants. Previous TILDA work has shown that, among other differences, respondents who did not attend the health assessment center had higher levels of physical disability, were weaker by grip strength, and had slower walking speed [64]. Appendix E provides further insights into the demographics of excluded and included participants. Notably, the excluded group had a higher percentage of deceased participants (24% vs. 10%) and was older (median age: 64 vs. 60 years). Differences in sex and educational attainment were observed, with self-reported diabetes being more prevalent among excluded participants (9% vs. 6%). Variations in factors like cardiovascular diseases and antihypertensive medication also suggest the potential for selection bias. Replication of these results in diverse populations is necessary to establish the external validity of the 3-item HI. Despite our efforts to control for confounding variables, residual confounding or unmeasured factors may still influence the observed associations. Also, the normalization process used for the 3-item HI relies on assumptions based on prior literature and may require further validation. Additionally, while our study employed Cox proportional hazards regression to analyze mortality risk, it is important to acknowledge the potential limitation of not exploring parametric models. While the underlying hazard function's smoothness might suggest a parametric approach, the Cox proportional hazards model's flexibility allowed us to capture complex and evolving risk dynamics without making strong assumptions about hazard distribution. Future investigations could consider parametric models to further assess the robustness of our findings and ascertain whether the observed associations persist under different modeling assumptions.’

Good luck

Comments on the Quality of English Language

Please do proofread the manuscript

Thank you, it has been done.

Submission Date

19 July 2023

Date of this review

09 Aug 2023 13:44:29

Please find below STARD checklist for this manuscript.

Section & Topic

No

Item

Reported on page #

TITLE OR ABSTRACT

1

Identification as a study of diagnostic accuracy using at least one measure of accuracy

(such as sensitivity, specificity, predictive values, or AUC)

7

ABSTRACT

2

Structured summary of study design, methods, results, and conclusions
(for specific guidance, see STARD for Abstracts)

1

INTRODUCTION

3

Scientific and clinical background, including the intended use and clinical role of the index test

1-3

4

Study objectives and hypotheses

3

METHODS

Study design

5

Whether data collection was planned before the index test and reference standard
were performed (prospective study) or after (retrospective study)

3

Participants

6

Eligibility criteria

3

7

On what basis potentially eligible participants were identified
(such as symptoms, results from previous tests, inclusion in registry)

3

8

Where and when potentially eligible participants were identified (setting, location and dates)

3

9

Whether participants formed a consecutive, random or convenience series

3

Test methods

10a

Index test, in sufficient detail to allow replication

3-7, 16-21

10b

Reference standard, in sufficient detail to allow replication

5

11

Rationale for choosing the reference standard (if alternatives exist)

N/A

12a

Definition of and rationale for test positivity cut-offs or result categories
of the index test, distinguishing pre-specified from exploratory

5-6

12b

Definition of and rationale for test positivity cut-offs or result categories
of the reference standard, distinguishing pre-specified from exploratory

5

13a

Whether clinical information and reference standard results were available
to the performers/readers of the index test

N/A

13b

Whether clinical information and index test results were available
to the assessors of the reference standard

N/A

Analysis

14

Methods for estimating or comparing measures of diagnostic accuracy

6-7

15

How indeterminate index test or reference standard results were handled

7

16

How missing data on the index test and reference standard were handled

7-8, 21-22

17

Any analyses of variability in diagnostic accuracy, distinguishing pre-specified from exploratory

6-7

18

Intended sample size and how it was determined

3, 7-8

RESULTS

Participants

19

Flow of participants, using a diagram

8

20

Baseline demographic and clinical characteristics of participants

8-9, 21-22

21a

Distribution of severity of disease in those with the target condition

8-9

21b

Distribution of alternative diagnoses in those without the target condition

8-9

22

Time interval and any clinical interventions between index test and reference standard

3, 5

Test results

23

Cross tabulation of the index test results (or their distribution)
by the results of the reference standard

8-9

24

Estimates of diagnostic accuracy and their precision (such as 95% confidence intervals)

9-13

25

Any adverse events from performing the index test or the reference standard

N/A

DISCUSSION

26

Study limitations, including sources of potential bias, statistical uncertainty, and generalisability

15

27

Implications for practice, including the intended use and clinical role of the index test

14-15

OTHER INFORMATION

28

Registration number and name of registry

16

29

Where the full study protocol can be accessed

16

30

Sources of funding and other support; role of funders

16